# An Error Compensation Method for Improving the Properties of a Digital Henon Map Based on the Generalized Mean Value Theorem of Differentiation

**DOI:** 10.3390/e23121628

**Published:** 2021-12-02

**Authors:** Yashuang Deng, Yuhui Shi

**Affiliations:** The School of Information and Safety Engineering, Zhongnan University of Economic and Law, Wuhan 430073, China; 202011200015@stu.zuel.edu.cn

**Keywords:** chaos, digital Henon map, dynamical degradation, error compensation, entropy

## Abstract

Continuous chaos may collapse in the digital world. This study proposes a method of error compensation for a two-dimensional digital system based on the generalized mean value theorem of differentiation that can restore the fundamental performance of chaotic systems. Different from other methods, the compensation sequence of our method comes from the chaotic system itself and can be applied to higher-dimensional digital chaotic systems. The experimental results show that the improved system is highly consistent with the real chaotic system, and it has excellent chaotic characteristics such as high complexity, randomness, and ergodicity.

## 1. Introduction

In nature and human society, a large number of interactions are non-linear. Chaos is an inherent characteristic of nonlinear dynamic systems and a common phenomenon in nonlinear systems. The chaos phenomenon is a deterministic and stochastic process that appears in a nonlinear dynamic system. This process is neither periodic nor convergent, and it has a sensitive dependence on the initial value. In daily life, behaviors such as population movement, weather changes, and social behavior can all be called chaotic [1]. Chaos promotes and relies on each other with other sciences. It is worth mentioning that, due to the special characteristics of chaos, such as initial sensitivity, ergodicity, unpredictability, and topological transitivity, a large number of security applications based on chaos have appeared [2].

The existing chaotic cipher schemes are mainly based on the chaotic system implemented on the digital precision equipment [3] and the digital chaotic system. However, in the implementation of a digital chaotic system, due to the limitations of limited precision equipment, the dynamic behavior of the digital chaotic system will degenerate, and the excellent characteristics of the chaotic system will disappear, such as the short cycle orbit, the low complexity, the strong correlation, and the uneven distribution, so that its output carries the characteristics of the system. If the attacker obtains part of the orbit information of the chaotic system and analyzes it according to the characteristics carried by the system, the chaotic system may be attacked. Therefore, the anti-degenerate chaotic system plays an important role in the field of cryptography and is of great significance to ensure the security of chaotic security systems.

At present, there are six methods to improve the dynamic degradation of digital chaotic systems. The first is to use high precision [4,5], which is the most direct method to improve power degradation. However, the cost of this method is high, and the effect achieved is not very good.The second is to cascade multiple chaotic systems. This method was proposed by Zhou et al. [6,7,8,9]. The main idea is to cascade two separate digital chaotic systems to increase the complexity of the digital chaotic system. This method is easy to implement and can make the output of the digital chaotic system not reflect any system characteristics, but it will lead to poor distribution. The third type is coupling [10,11,12,13,14,15,16,17,18,19], which couples the digital chaotic system and the continuous chaotic system one way or two ways. This method cleverly makes use of the complementarity between the digital chaotic system and the continuous chaotic system, but, compared with other methods, this method is more expensive to implement, and the system is more complex. The fourth type is switching [20], which means that when the digital chaotic system is about to appear degraded, the digital chaotic system is switched to another digital chaotic system, so as to avoid the short period of the system. The effect of this method largely depends on the rules of switching. The fifth type is perturbation [21,22,23,24,25,26,27,28,29,30]. Perturbation refers to adding a slight disturbance to the system. When the system is about to enter a cycle, the state can be pulled away immediately to prevent it from entering a cycle. However, the effect depends largely on the choice of the perturbation source. The sixth is the error-compensation method [31]. The purpose of this method is to restore the original chaotic system as much as possible, compensating the truncated part of the digital system. This method is less expensive to implement, and the effect is more obvious. Therefore, this study adopted an error-compensation method to improve the performance of the digital chaotic system.

Hu et al. [31] proposed a variable parameter compensation method (VPCM) based on Lyapunov exponents. They used variable parameters and Lyapunov exponents to improve the performance of logistic digital systems according to the differential median theorem. Deng et al. [16] made further improvements on the basis of the former and proposed a logistic digital system control method (VPCMDP) based on the differential mean theorem and state feedback technology. Both of these two methods are based on error compensation. The purpose of the former is to restore the digital logistic system to the real chaotic logistic chaotic system, and the improvement of the latter further combines the digital logistic system with cryptographic applications. However, the disadvantage of these two methods is that they cannot be applied to high-dimensional systems. According to [32], they extended the differential median theorem of one-dimensional functions to multi-dimensional functions. Therefore, we proposed a method of error compensation for a two-dimensional digital system (Henon map) based on the generalized mean value theorem of differentiation to make it close to the real chaotic system. Additionally, we named our method as the error-compensation method for high-dimensional systems (ECMHD).

The second section of this article introduces the two-dimensional chaotic system-Henon mapping and expands the description of its dynamic degradation behavior when it is implemented on digital devices; the third section proposes a new compensation method ECMHD for the digital Henon system; the fourth section compares it with the digital Henon system and the other three latest methods; and the last section is the conclusion of this article.

## 2. Dynamical Degradation of the Digital Henon Map

### 2.1. Henon Map

A Henon map is a discrete-time dynamic system that can produce chaotic phenomena. Its iterative expression is shown as follows:(1)$$\left\{\begin{array}{c} x_{i+1}=1-a{x_{i}}^{2}+y_{i}\\ y_{i+1}=bx_{i} \end{array}\right.$$
where $x_{i}\in \left[-1.5,1.5\right]$, $y_{i}\in \left[-0.4,0.4\right]$, a, and b are parameters. When *a* = 1.4 and *b* = 0.3, it is Henon’s typical mapping. At this time, the mapping is chaotic. Under atypical values, this mapping may be non-chaotic and may exhibit periodicity. Theoretically, the two-dimensional Henon map has good ergodicity and random orbits in the phase space. However, when the Henon map is implemented on a finite precision device, its performance will degrade and no longer show chaotic phenomena. It will degenerate into the following digital system:(2)$$\left\{\begin{array}{c} x_{i+1}^{*}=1-ax{{}_{i}^{*}}^{2}+by_{i}^{*}\\ y_{i+1}^{*}=bx_{i}^{*} \end{array}\right.$$
where $x_{i}^{*}=FL\left(x_{i}\right)$; $y_{i}^{*}=FL\left(y_{i}\right)$; f is a non-linear function; and FL(·) is a quantization function, which is $FL\left(x\right)=\left\lfloor x\cdot 2^{p}\right\rfloor \cdot 2^{-p}$. It represents rounding down and returns the integer that is less than or equal to the function parameter closest to it. We set a low precision *p* = 8; the lower the precision, the lower the cost of system implementation, and it can also prove the effectiveness of our method. We randomly chose the initial values $x_{0}\left(1\right)=0.9649$ and $y_{0}\left(1\right)=0.1576$; the parameters were *a* = 1.4 and *b* = 0.3. Then, we observed the digital Henon map with degraded dynamics.

### 2.2. Dynamical Degradation of Digital Henon Map

We analyzed the trajectory of the digital Henon system; the trajectory can be the most intuitive to see the state of the system. Take the x-dimension as an example. It can be seen from Figure 1a that the digital Henon system has a period-doubling phenomenon after about 50 iterations. Then, we analyzed the state frequency distribution of the system. It can be seen from Figure 1b that the system mainly focuses on the states of several phase diagrams. This is because the state of the digital Henon system is limited to $\Omega_{p}=\left\{x_{i}=k\times 2^{-p}|k=0,1,2,\ldots ,2^{p-1}\right\}$. Then, the phase diagram of the system, as shown in Figure 1c, will not achieve ergodicity. At the same time, we analyzed the correlation of the system: auto-correlation, also called sequence correlation, is the cross-correlation of a signal with itself at different points in time, which is a good measure of randomness. It can be seen from Figure 1d that the auto-correlation of the system was relatively strong. Cross-correlation means the degree of correlation between two time series. Here, we made small changes to the initial value, $x_{0}\left(2\right)=0.9649+2^{-p}$, $y_{0}\left(2\right)=0.1576+2^{-p}$, and we obtained a new sequence. We tested the correlation between the newly obtained sequence and the previous sequence, and the result is shown in Figure 1e, which shows that the two sequences have a high correlation. Figure 1f shows the signal power spectrum, which did not have ultra-wide band requirements.

In addition, we also used approximate entropy and information entropy to measure the complexity of the digital Henon map. Approximate entropy is a nonlinear dynamic parameter used to quantify the regularity and unpredictability of time series fluctuations. It uses a non-negative number to represent the complexity of a time series and reflects the possibility of new information in the time series. The more complex the time series, the greater the approximate entropy. We measured the approximate entropy of the system under different initial values when *p* = 8. As shown in Figure 2a, the approximate entropy of the digital Henon system was stable at 0.2–0.3, which shows that the complexity of the system was relatively low. Figure 2b shows the approximate entropy under the same initial value and different precision. It can be seen that when the precision is low, the approximate entropy of the digital system is positively correlated with the precision of the digital Henon system. When the precision reaches 24, it basically stabilizes at 0.7. Information entropy is a parameter that measures the uncertainty or unpredictability of a time series. When *p* = 8, the ideal value of information entropy is 8. As shown in Figure 2c, the information entropy under different initial values is stable at about 5, indicating that the system has strong regularity.

## 3. A Novel Method for a Digital Henon Map

### 3.1. Method Description

From Equations (Equation 1) and (Equation 2), it was concluded that there were errors between $x_{i}$ and $x_{i}^{*}$ and $y_{i}$ and $y_{i}^{*}$. We used $\Delta x_{i}$ and $\Delta y_{i}$ to represent them, respectively. The relationship between them can be expressed by Equations (Equation 3) and (Equation 4)
(3)$$x_{i}=x_{i}^{*}+\Delta x_{i}$$
(4)$$y_{i}=y_{i}^{*}+\Delta y_{i}$$

According to the one-dimensional differential mean value theorem:(5)$$\begin{array}{cc} f\left(x+\Delta x\right)=f\left(x\right)+f^{\prime }\left(x+\theta \Delta x\right)\cdot \Delta x & 0<\theta <1 \end{array}$$

We get Equation (Equation 6)
(6)$$\begin{array}{cc} f\left(x_{i}\right)=f\left(x_{i}^{*}+\Delta x_{i}\right)=f\left(x_{i}^{*}\right)+f^{\prime }\left(x_{i}^{*}+\theta \Delta x_{i}\right)\Delta x_{i} & 0<\theta <1 \end{array}$$

Let $\varepsilon =x_{i}^{*}+\theta \left(x_{i}-x_{i}^{*}\right)$. Since the Henon map is a high-dimensional map, according to some one-dimensional theorems provided by the [32], it can be extended to high-dimensional space examples. We list its Jacobian matrix: (7)$$\left[\begin{array}{c} f(x_{i})\\ f(y_{i}) \end{array}\right]=\left[\begin{array}{c} f(x_{i}^{*})\\ f(y_{i}^{*}) \end{array}\right]+\left[\begin{array}{cc} \frac{\partial x}{\partial f(x)} & \frac{\partial y}{\partial f(x)}\\ \frac{\partial x}{\partial f(y)} & \frac{\partial y}{\partial f(y)} \end{array}\right]\cdot \left[\begin{array}{c} x_{i}-x_{i}^{*}\\ y_{i}-y_{i}^{*} \end{array}\right]=\left[\begin{array}{c} f(x_{i}^{*})\\ f(y_{i}^{*}) \end{array}\right]+\left[\begin{array}{cc} -2a\varepsilon & 1\\ b & 0 \end{array}\right]\cdot \left[\begin{array}{c} x_{i}-x_{i}^{*}\\ y_{i}-y_{i}^{*} \end{array}\right]$$

We obtained the new compensated Henon map as Equation (Equation 8) and the architecture of ECMHD for the digital Henon map is shown in Figure 3:(8)$$\left\{\begin{array}{cc} x_{i+1}=f\left(x_{i}^{*}\right)-2a\varepsilon \left(x_{i}-x_{i}^{*}\right)+y_{i}-y_{i}^{*} & \\ y_{i+1}=f\left(y_{i}^{*}\right)+b\left(x_{i}-x_{i}^{*}\right) & \\ \varepsilon =x_{i}^{*}+\theta \left(x_{i}-x_{i}^{*}\right) & 0<\theta <1 \end{array}\right.$$

### 3.2. Performance Analysis

As in the second section, we set $x_{0}\left(1\right)=0.9649$ and $y_{0}\left(1\right)=0.1576$, and the parameters were *a* = 1.4 and *b* = 0.3. Theta can take any value between 0 and 1. We arbitrarily set theta = 0.5 for the performance analysis.

Figure 4a shows the trajectory of the system after error compensation. It can be seen from the figure that whenever the system is about to enter the cycle, a slight perturbation causes the system to jump out and the cycle is prolonged. It can also be seen from the distribution in Figure 4b that the improved system has achieved ergodicity, and the chaotic state has a wide range, similar to the distribution diagram of the real chaotic system.We can also observe from Figure 4c that the phase diagram of the chaotic system is composed of multiple horizontal parabolas, which is highly consistent with the phase diagram of the real chaotic system. From Figure 4d,e, we can see that the correlation of the improved chaotic system has been greatly improved. As shown in Figure 4d, the auto-correlation is similar to an impulse function, and the cross-correlation is Almost 0, indicating that the improved system has strong randomness and weak correlation. Due to the enhanced chaotic performance of the improved system, its signal power spectrum also presents a wide-screen band mode, as shown in Figure 4f.

From the above analysis, the Henon system after error compensation was well improved in terms of randomness, periodicity, and ergodicity.

## 4. Dynamical Performance Comparison

### 4.1. Performance Comparison of the Systems before and after Error Compensation

In this section, we compare the improved system with the digital Henon system. We set the precision to *p* = 8; the initial value to $x_{0}\left(1\right)=0.9649$ and $y_{0}\left(1\right)=0.1576$; and the parameters to *a* = 1.4 and *b* = 0.3.

#### 4.1.1. Trajectories and Period

We first compare from the most intuitive trajectory diagram. Figure 5a,b show the trajectories of Equation (Equation 2) and Equation (Equation 8), respectively. It is not difficult to see that Figure 5a entered the cycle when iterated to about step 50. In Figure 5b, before the system entered the cycle, the state was pulled out of the cycle by a slight error compensation. So, it can be seen from the figure that the perturbed Henon system is still random. This shows that the ECMHD effectively prolongs the cycle of the digital system. In order to observe the periods of the two systems more carefully, we measured the specific periods of the two systems at low precision and the number of steps that they took to enter the period for the first time, as shown in Table 1. From the Table 1, we can see that when *p* = 8, the digital system enters the cycle of cycle 23 when it is iterated to the 54th time. As the precision increases, the period and the number of iteration steps into the cycle also increase with the precision and become longer. The cycle of a system that has been improved by error compensation has been greatly extended, and the period can no longer be predicted only when the precision is 10. It can be seen that the short-period phenomenon can be eliminated after the error compensation.

#### 4.1.2. Frequency Distribution and Phase Diagram

Figure 6a,b are the distributions of Equations (Equation 2) and (Equation 8), respectively. Figure 6c,d are the phase diagrams of Equations (Equation 2) and (Equation 8), respectively. It can be seen that the phase diagram of the system after error compensation was much denser than the phase space mapped by the digital Henon map.The distribution was not only concentrated in several states, which shows that the error compensation method can greatly improve its ergodicity, and the phase diagram of the perturbed system was very similar to the real chaotic system, which also shows that the ECMHD still maintained the original system structure.

#### 4.1.3. Correlation

After error compensation, the strong auto-correlation in Figure 7a was strengthened into the auto-correlation of an ideal chaotic sequence. As shown in Figure 7b, the auto-correlation function will disappear rapidly with the interval. This is similar to trigonometric functions. Its cross-correlation also changed from the high correlation in the large interval of the digital system (Figure 7c) to a cross-correlation function close to zero (Figure 7d). This fully shows that the method effectively improves the chaotic characteristics of the digital system.

#### 4.1.4. Entropy

We compared the approximate entropy under different precisions. It can be seen from Figure 8 that when the precision was very low, the approximate entropy of the improved system was twice that of the digital system. Only when the precision reached 20 was the difference between the two systems very small. Additionally, the approximate entropy of the ECMHD was always stable. In order to verify that the performance of the improved system was not affected by the initial value, we also compared it under different initial values. As shown in Figure 9, it is not difficult to see that the approximate entropy of the compensated system was much higher than that of the system number.

We also compared the information entropy. As shown in Figure 10, under *p* = 8 and different initial values, the information entropy of the compensated system was very close to the ideal value of 8, which was much higher than the information entropy of the digital system. This shows that the randomness of the improved system was also significantly improved, close to the ideal value.

### 4.2. Comparison of the Proposed Error Compensation Scheme with Existing Methods

In this section, we compare ECMHD with the other three latest methods. The first one is from Liu et al. [30]. Their method is that in the iterative process of the digital system, as long as there are repeated states, the parameters and states of the system will be disturbed, so that the system jumps out of the cycle. The second one is from Wu et al. [31]. They constructed a control function by introducing the iteration time to replace the control parameters in the original chaotic map. The third one is from Tang et al. [33]. They introduced delayed state variables in the digital system and used the state variables of one system to change the control parameters of another system. We compared these methods under the condition of *p* = 8.

#### 4.2.1. Trajectories

Figure 11 shows the trajectory of the latest several improvement methods and the ECMHD. It can be seen intuitively that when Liu’s method iterated to about 100, the system entered a cycle. Wu’s and Tang’s method was that the system has a cycle after walking more than 50 steps. Our method showed better performance without periodic phenomena. This shows that the ECMHD can extend the cycle of the system well even with low precision.

#### 4.2.2. Frequency Distribution

It can be seen from the Figure 12 that the ECMHD can be highly consistent with the distribution of the real chaotic system even at lower precision. This shows that the ECMHD is not restricted to low-precision states. The other several improvement methods expose their limited state range under low precision. Their improved method is still affected by dynamic degradation at low precision and cannot achieve ergodicity. It shows that the ECMHD is more capable of resisting attacks than other methods.

#### 4.2.3. Auto-Correlation

It can be seen from the Figure 13 that their improvement method had a greater improvement with the digital system when the accuracy was lower. However, their auto-correlation function still maintained a high correlation in a large range. The auto-correlation function of the ECMHD decreases rapidly with the passage of time. Then, it remains stable around 0 to form an impulse function. This shows that the ECMHD is more random.

#### 4.2.4. Entropy

In order to further compare the performance of these methods, we analyzed and compared their entropy. We first tested the approximate entropy of different initial values with *p* = 8. It can be clearly seen from Figure 14 that the approximate entropy of the ECMHD is far superior to other methods, and it was stable at around 0.7. The approximate entropy of Liu’s method was stable around 0.1. Wu’s method was stable at around 0.4. Tang’s method fluctuated between 0.2–0.5. This fully shows that the ECMHD has higher complexity and is more unpredictable.

Figure 15 shows the information entropy under the *p* = 8 and different initial values. It can be seen that the ECMHD was very close to the ideal value of 8. However, the other methods are still far from the ideal value. Moreover, Tang’s method also showed relatively low and unstable information entropy values. This means that the ECMHD has greater uncertainty than the other systems.

#### 4.2.5. Similarity Comparison with Real Chaotic System

There are two basic categories of similarity. The first one is objective similarity. It means that the similarity between objects is a certain functional relationship between the multi-dimensional features of the objects, such as the Euclidean distance between objects. The second one is subjective similarity. It means that the similarity is the cognitive relationship between people and the research object, which depends on the person and the environment in which they are located. The subjective similarity meets the visual needs of the human eye and has a certain degree of ambiguity.

In order to judge the similarity with the original system more accurately, we used the Euclidean distance for measurement here. The Euclidean distance is the most common representation of the distance between two or more points, also known as the Euclidean metric, which is defined in Euclidean space. The distance between $x=(x_{1},...,x_{n})$ and $y=(y_{1},...,y_{n})$ is defined as Equation (Equation 9):(9)$$d\left(x,y\right)=\sqrt{{\left(x_{1}-y_{1}\right)}^{2}+{\left(x_{2}-y_{2}\right)}^{2}+...+{\left(x_{n}-y_{n}\right)}^{2}}=\sqrt{\sum_{i=1}^{n}{\left(x_{i}-y_{i}\right)}^{2}}$$

According to the above Equation (Equation 9), we calculated the distance between each method and its real chaotic system as shown in Table 2:

It can be seen that the ECMHD was more consistent with the real chaotic system, which shows that our compensation effect is better.

Through the above analysis, it is concluded that, under severe conditions, the other three methods still cannot avoid the collapse of the chaotic system in the digital world, and they still showed obvious cycles and limited state space while being unable to be traversed and showing a strong correlation. For other problems, the AE and IE values were also far lower than the ideal value; so, there is still a large gap in the agreement with the original chaotic system. The ECMHD can still maintain the chaotic characteristics, which also shows that the ECMHD has high performance in reality while ensuring low cost; so, it can be more widely used.

## 5. Conclusions

This study proposed a method of error compensation for a two-dimensional digital system based on the generalized mean value theorem of differentiation. The proposed method, ECHMD, can not only make the digital Henon system behave chaotically and show the ideal characteristics again, including ergodicity, higher complexity, an auto-correlation similar to δ, and close-to-zero cross-correlation, but it is also highly consistent with the original real Henon system. The study also compared the performance of three recent methods. Under the same severe conditions, we compare and analyze several other methods from the perspectives of period, state distribution, autocorrelation, approximate entropy, and information entropy. This shows that the ECMHD can achieve a low-cost, high-performance effect. In the future, we will add an adaptive control to the improved system to further optimize system performance.

## Figures and Tables

**Figure 1 entropy-23-01628-f001:**
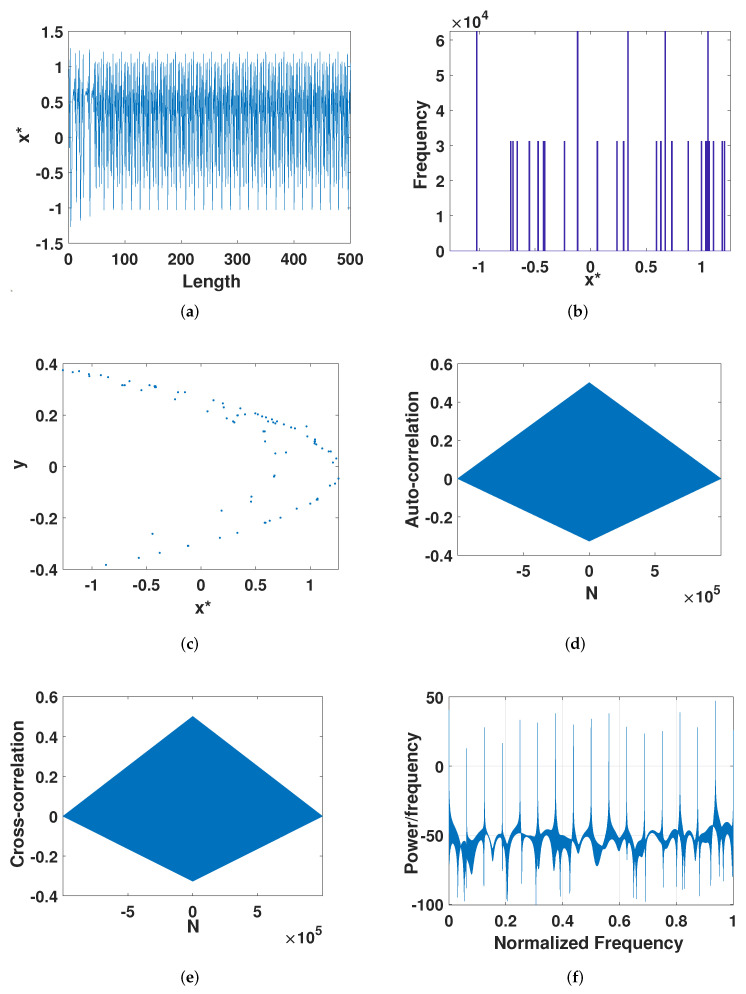
The dynamical properties of the digital Henon map. (**a**) The *x*-dimensional trajectory. (**b**) The *x*-dimensional distribution. (**c**) Phase diagram. (**d**) Auto-correlation. (**e**) Cross-correlation. (**f**) Signal power spectrum.

**Figure 2 entropy-23-01628-f002:**
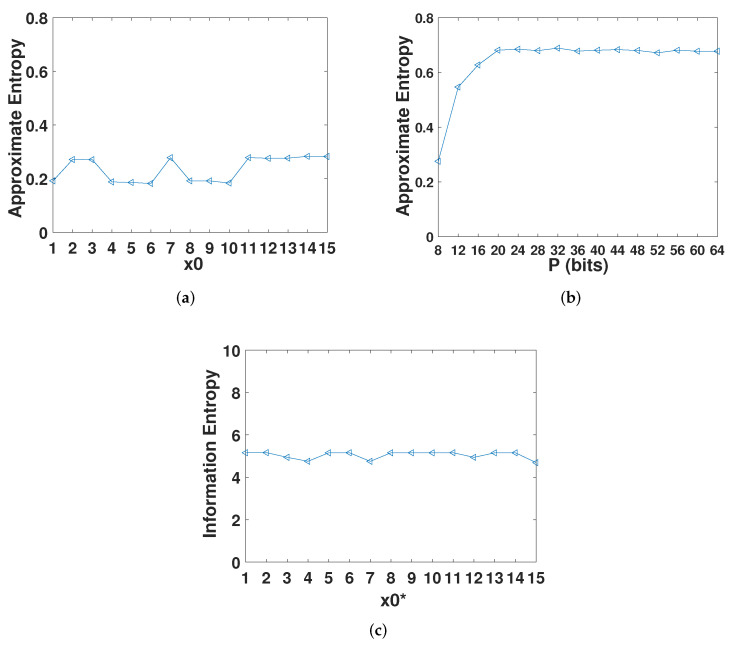
Entropy analysis of the digital Henon map. (**a**) Approximate entropy with different initial value. (**b**) Approximate entropy under different precision. (**c**) Information entropy with different initial value.

**Figure 3 entropy-23-01628-f003:**
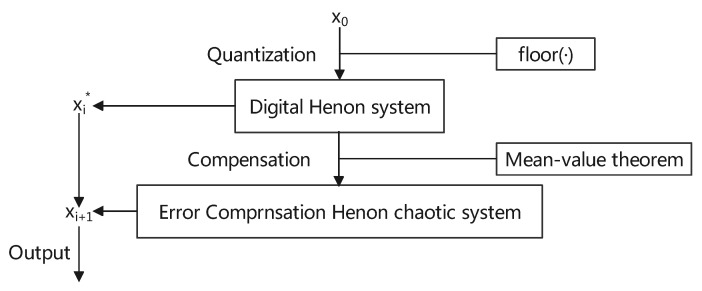
Architecture of ECMHD for the digital Henon map.

**Figure 4 entropy-23-01628-f004:**
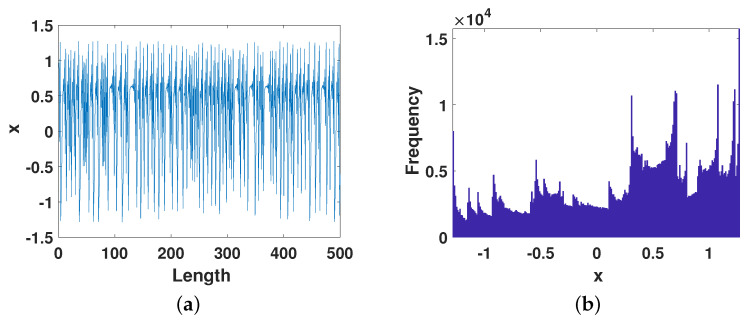
The dynamical properties of the digita Henon map after error compensation. (**a**) The *x*-dimensional trajectory. (**b**) The *x*-dimensional distribution. (**c**) Phase diagram. (**d**) Auto-correlation. (**e**) Cross-correlation. (**f**) Signal power spectrum.

**Figure 5 entropy-23-01628-f005:**
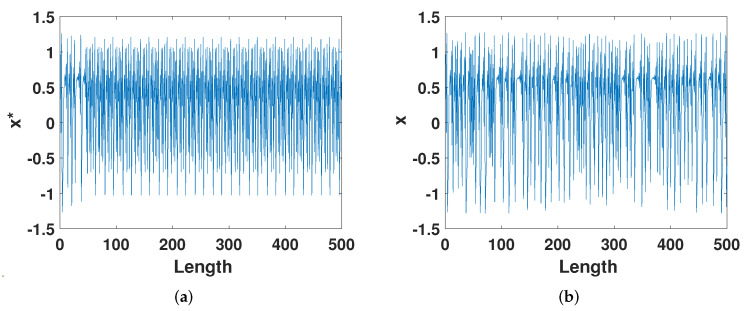
(**a**) The trajectory of the digital Henon map. (**b**) The trajectory of the improved Henon map.

**Figure 6 entropy-23-01628-f006:**
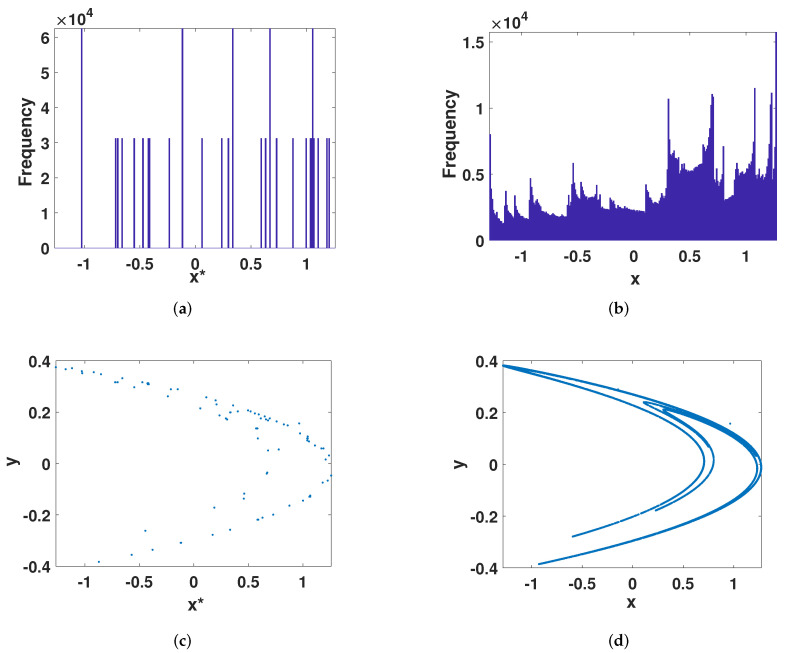
Frequency distribution and phase diagram. (**a**) The frequency distribution of the digital Henon map. (**b**) The frequency distribution of the improved Henon map. (**c**) The phase diagram of the digital Henon map. (**d**) The phase diagram of the improved Henon map.

**Figure 7 entropy-23-01628-f007:**
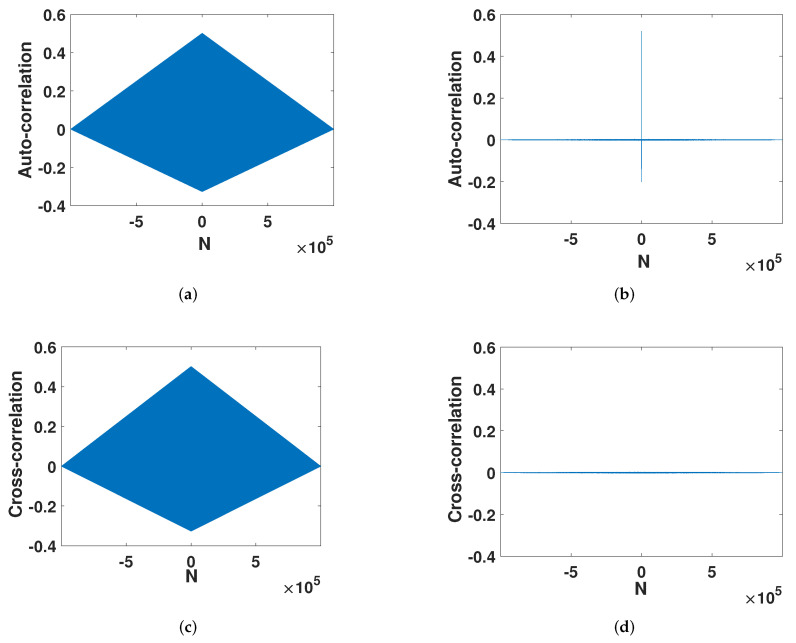
Correlation. (**a**) The auto-correlation of the digital Henon map. (**b**) The auto-correlation of the improved Henon map. (**c**) The cross-correlation of the digital Henon map. (**d**) The cross-correlation of he improved Henon map.

**Figure 8 entropy-23-01628-f008:**
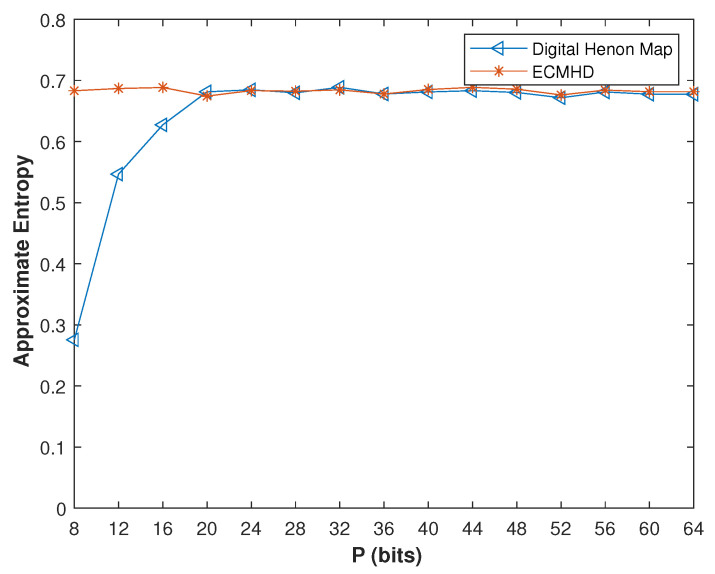
The approximate entropy of the systems before and after error compensation under different precisions.

**Figure 9 entropy-23-01628-f009:**
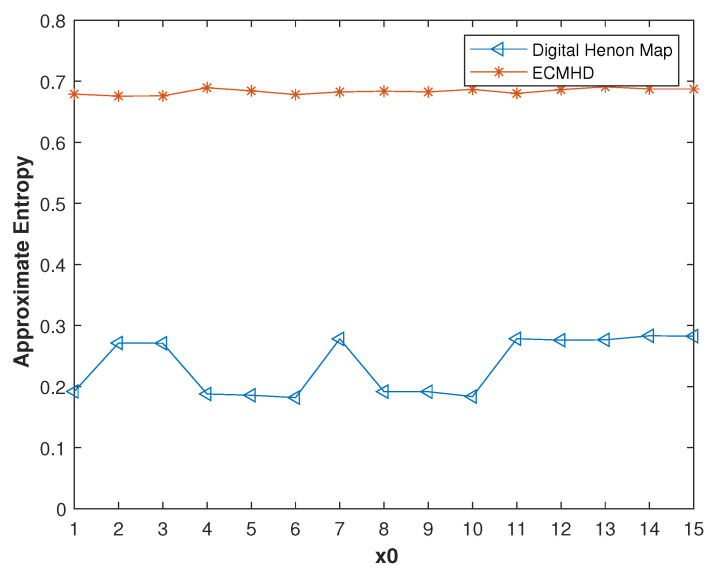
The approximate entropy of the systems before and after error compensation with different initial values.

**Figure 10 entropy-23-01628-f010:**
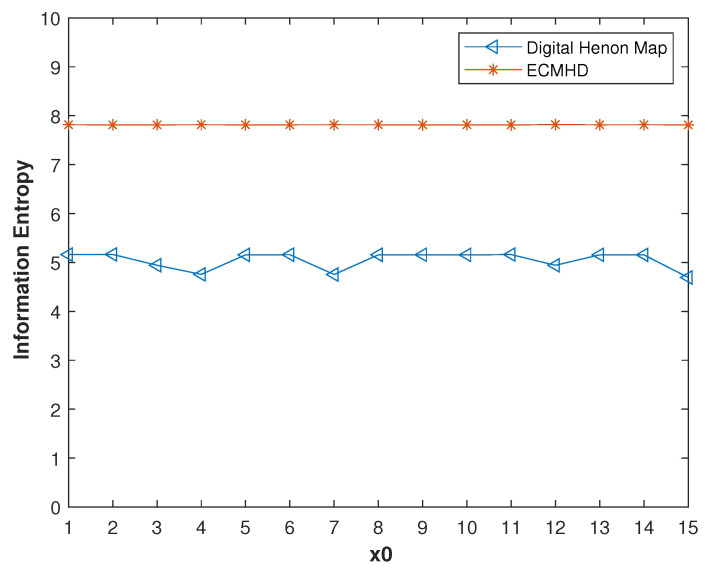
The information entropy of the systems before and after error compensation with different in initial values.

**Figure 11 entropy-23-01628-f011:**
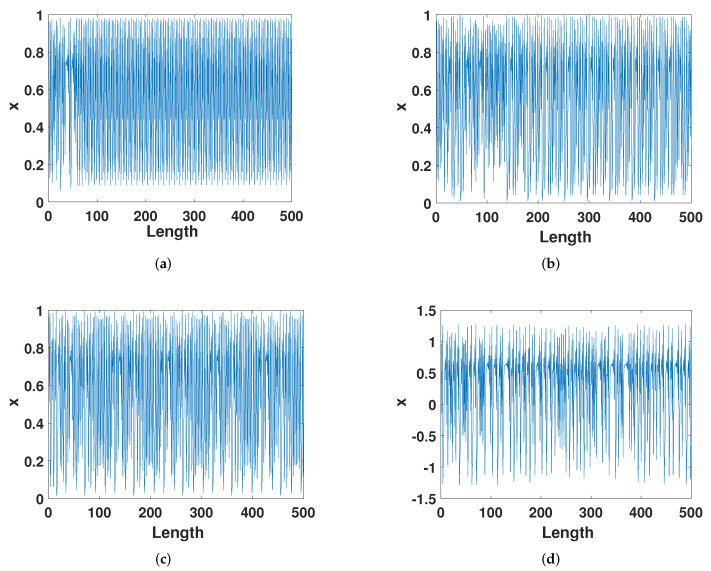
The trajectory of different methods. (**a**) The trajectory of Liu’s method. (**b**) The trajectory of Wu’s method. (**c**) The trajectory of Tang’s method. (**d**) The trajectory of ECMHD.

**Figure 12 entropy-23-01628-f012:**
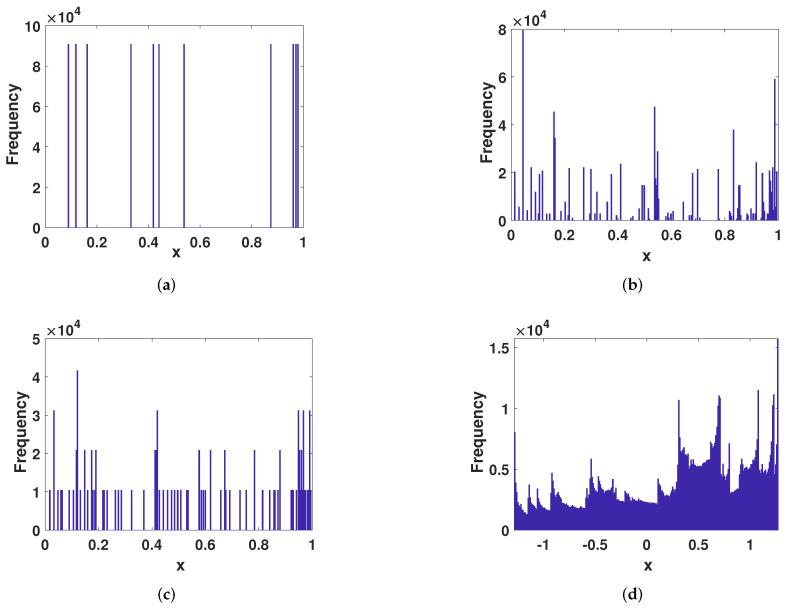
The frequency distribution of different methods. (**a**) The frequency distribution of Liu’s method. (**b**) The frequency distribution of Wu’s method. (**c**) The frequency distribution of Tang’s method. (**d**) The frequency distribution of ECMHD.

**Figure 13 entropy-23-01628-f013:**
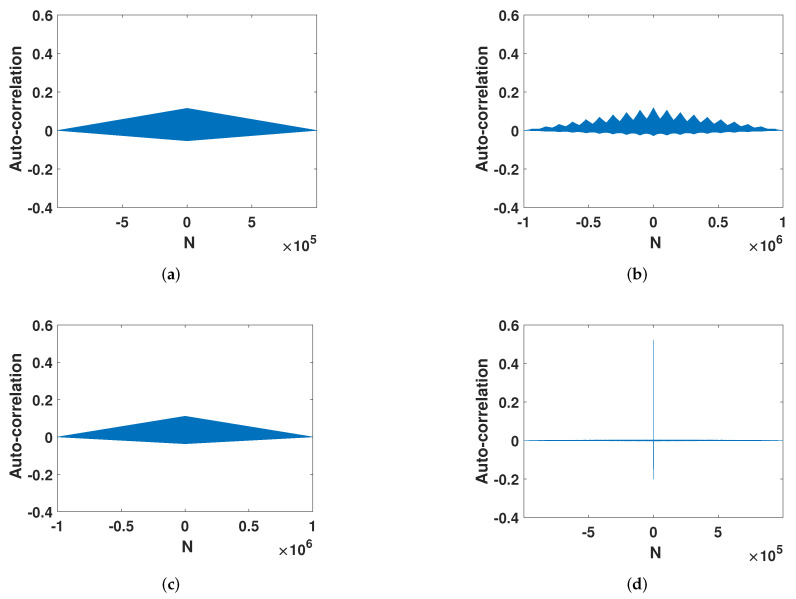
The auto-correlation of different methods. (**a**) The auto-correlation of Liu’s method. (**b**) The auto-correlation of Wu’s method. (**c**)The auto-correlation of Tang’s method. (**d**) The auto-correlation of the ECMHD.

**Figure 14 entropy-23-01628-f014:**
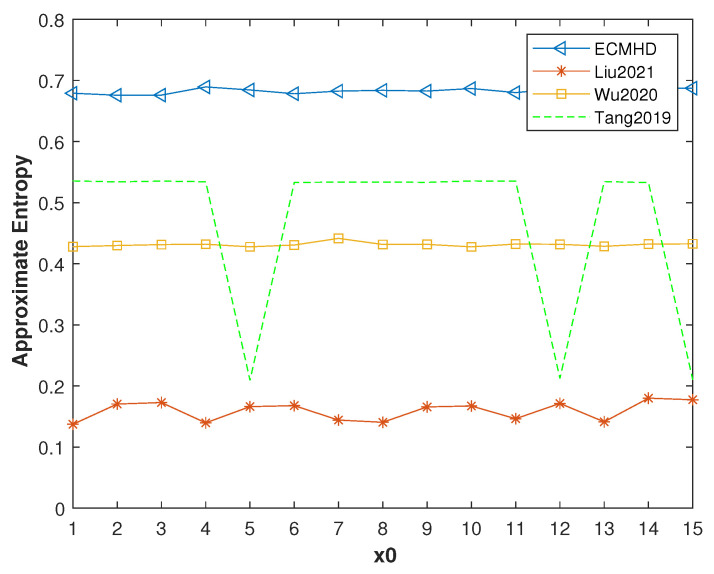
Approximate entropy values of improved systems for different methods.

**Figure 15 entropy-23-01628-f015:**
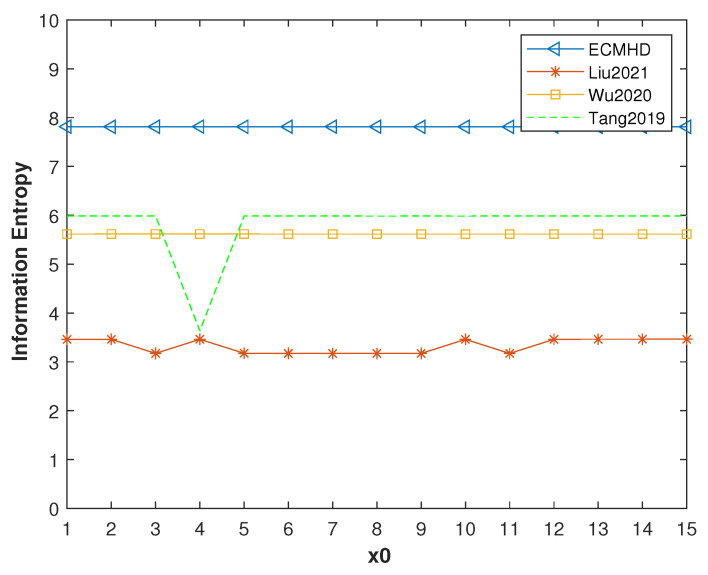
Information entropy values of improved systems for different methods.

**Table 1 entropy-23-01628-t001:** Period of the digital Henon map before and after error compensation.

Precision	Period (Digital)	Period (ECMHD)	The Place It Fell into Cycle (Digital)	The Place It Fell into Cycle (ECMHD)
8	2323	83,690	54	196,964
9	252	217,763	26	6179
10	193	U	51	U
11	194	U	209	U
12	233	U	507	U
13	1295	U	234	U
14	891	U	88	U
15	1849	U	540	U
16	1364	U	732	U
17	2134	U	857	U
18	3914	U	4721	U
19	3359	U	6551	U
20	10,747	U	10,945	U
21	13,885	U	9344	U
22	754	U	17,357	U
23	10,031	U	72,207	U
24	32,513	U	59,022	U

**Table 2 entropy-23-01628-t002:** The distance from the original system.

Method	Distance
Liu’s method	455.9403
Wu’s method	459.8772
Tang’s method	636.9570
ECMHD	252.1862

## Data Availability

Data sharing is not applicable.

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
