# Peer review of "An Error Compensation Method for Improving the Properties of a Digital Henon Map Based on the Generalized Mean Value Theorem of Differentiation"

_entropy, 2021, doi:10.3390/e23121628_

Round 1
Reviewer 1 Report
The paper “An Error Compensation Method of Improving the Properties of Digital Henon Map Based on Generalized Mean Value Theorem of Differentiation” proposes a method of error compensation for a two-dimensional digital system based on generalized mean value theorem of differentiation. It is of interest and novelty, so I suggest considering it for publication after minor changes:
- A graphical abstract would add interest to catch the eye
- The abstract correctly states the activities carried out but should also include the novelty, a brief intro, and objectives of the study. Please supplement it.
- In the text the term our method is often used, it could be replaced by a more impersonal form, perhaps an acronym, for later reference from other papers.
- Conclusion includes personal impressions of the researchers. “We found that the improved effect is very good and highly consistent with the original real henon system, and has good chaotic characteristics.” could you please change it for a more unpersonal way of expressing it.
- Conclusions with quantitative results could also be added
Reviewer 2 Report
The manuscript has proposed a method of error compensation for two dimensional digital system. It is an impressive case study with a well documented format. However, the authors can think of describing the detailed information and comparison with the other state-of-the-art methods, which are already discussed in the manuscript. This comparison can provide a great insight of the topic to the readers.
Besides, the authors must discuss the motivation of the study in detail.
